# Somatic clones heterozygous for recessive disease alleles of *BMPR1A* exhibit unexpected phenotypes in *Drosophila*

**Takuya Akiyama[1], Sırma D User[1], Matthew C Gibson[1,2]***

[1]Stowers Institute for Medical Research, Kansas City, United States; [2]Department of Anatomy and Cell Biology, The University of Kansas School of Medicine, Kansas City, United States

**Abstract** The majority of mutations studied in animal models are designated as recessive based on the absence of visible phenotypes in germline heterozygotes. Accordingly, genetic studies primarily rely on homozygous loss-of-function to determine gene requirements, and a conceptually-related 'two-hit model' remains the central paradigm in cancer genetics. Here we investigate pathogenesis due to somatic mutation in epithelial tissues, a process that predominantly generates heterozygous cell clones. To study somatic mutation in *Drosophila*, we generated inducible alleles that mimic human Juvenile polyposis-associated *BMPR1A* mutations. Unexpectedly, four of these mutations had no phenotype in heterozygous carriers but exhibited clear tissue-level effects when present in somatic clones of heterozygous cells. We conclude that these alleles are indeed recessive when present in the germline, but nevertheless deleterious when present in heterozygous clones. This unforeseen effect, *deleterious heteromosaicism*, suggests a 'one-hit' mechanism for disease initiation that may explain some instances of pathogenesis associated with spontaneous mutation.

DOI: https://doi.org/10.7554/eLife.35258.001

***For correspondence:**
mg2@stowers.org

**Competing interests:** The authors declare that no competing interests exist.

## Introduction

Genomes are inherently unstable, an attribute that drives evolution at the population level but simultaneously underlies the incidence of spontaneous disease (*Veltman and Brunner, 2012*; *Frank, 2014*; *Campbell et al., 2015*; *Acuna-Hidalgo et al., 2016*; *Forsberg et al., 2017*; *Stenson et al., 2017*). In humans, DNA replication error is estimated to induce mutations at a frequency of $10^{-9}$ per replicative cycle such that newborns already exhibit substantial somatic mosaicism at birth (*Lynch, 2010*; *Frank, 2014*; *Campbell et al., 2015*; *Fernández et al., 2016*; *Forsberg et al., 2017*; *Machiela and Chanock, 2017*). Additional somatic mutations accumulate over time, resulting in the age-dependent incidence of various pathologies, including cancer, X-linked disorders, and neurodevelopmental disease (*Veltman and Brunner, 2012*; *Deng et al., 2014*; *Frank, 2014*; *Fernández et al., 2016*; *Machiela and Chanock, 2017*). Indeed, it is estimated that spontaneous DNA replication error accounts for two-thirds of cancer mutations (*Tomasetti et al., 2017*). Nevertheless, little is known about the underlying cellular mechanisms by which somatic mutations trigger disease, primarily because of the technical difficulty of inducing specific mutations in endogenous loci during development of model animals.

## Results and discussion

To investigate the phenotypic consequences of disease-associated alleles in vivo, previous studies have frequently employed methods that drive aberrant protein expression under the control of non-endogenous promoter elements. The *Drosophila* Gal4/UAS system, for example, provides a highly flexible platform to drive gene expression in vivo (*Brand and Perrimon, 1993*). This methodology can effectively model tissue mosaicism for putative disease alleles, but results can be difficult to interpret due to the reliance on gene overexpression through exogenous regulatory elements. Indeed, even *wild-type* alleles expressed at non-physiological levels can produce aberrant phenotypes (*Figure 1—figure supplement 1*) (*Rørth, 1996*; *Grieder et al., 2007*; *Akiyama et al., 2008*).

To more rigorously investigate the effects of disease-associated mutations in vivo, we used CRISPR/Cas9-dependent genome editing to establish a Flippase (FLP) recombination-dependent allele switch system. Using human disease-associated alleles of the *Drosophila Bone Morphogenetic Protein Receptor 1A (BMPR1A)* homologue *thickveins* (*tkv*) as a model, our approach permits the inducible conversion of the *wild-type* locus to a specific mutant allele (*Figure 1A*; *Figure 1—figure supplement 2A*). In *Drosophila*, signaling through BMPR1A/Tkv is required to control growth and direct cell fates in the developing wing imaginal disc. The primary ligand, Decapentaplegic (DPP), is expressed in a narrow band of cells at the anterior-posterior compartment boundary, leading to a downstream phospho-Mothers against dpp (p-Mad) activity gradient in the center of the disc that functions to establish presumptive wing vein regions (*Figure 1B–D*) (*Lecuit et al., 1996*; *Nellen et al., 1996*; *Affolter and Basler, 2007*; *Restrepo et al., 2014*; *Akiyama and Gibson, 2015b*; *Akiyama and Gibson, 2015a*).

We first engineered a switchable control allele, $tkv^{GFP>>mCherry}$, which carries tandem duplications of the *wild-type tkv* coding sequence. Each *tkv* copy is tagged with a distinctive fluorophore such that Green Fluorescent Protein (GFP) labels the 5' allele and mCherry labels the 3' copy. *FRT* sites flank the *tkv-GFP* coding exons to allow recombinogenic removal of *tkv-GFP* and thus a switch to expression of *tkv-mCherry* within the endogenous regulatory architecture (*Figure 1A*; *Figure 1—figure supplement 2A*). Under normal conditions, transcription terminates before the *tkv-mCherry* coding sequence and only Tkv-GFP is expressed (*Figure 1B–D*; *Figure 1—figure supplement 2B*). Marked allele switch events can thus be induced in precise spatio-temporal patterns using controlled FLP expression to excise *tkv-GFP* (*Figure 1E–G*; *Figure 1—figure supplement 2C–E*). In control experiments, we detected no adverse effects of $tkv^{GFP>>mCherry}$ (*green*) conversion to $tkv^{>mCherry}$ (*red*). In fact, while *UAS-tkv* overexpression in the wing pouch using *nub-Gal4* caused abnormal phenotypes (*Figure 1—figure supplement 1*), induction of Tkv-mCherry in the *nub-Gal4* domain had no detectable effect on development (*Figure 1B–G*).

To further validate the allele switch methodology, we generated a transgenic line that permits inducible in-locus expression of $Tkv^{Q199D}$, a constitutively active form of the receptor (*Hoodless et al., 1996*). As expected, *w; $tkv^{GFP>>Q199D-mCherry}$/nub-Gal4; UAS-FLP/+* wing discs exhibited ectopic BMP/DPP activity throughout the Nub expression domain, resulting in both wing disc and adult wing abnormalities (*Figure 1H–J*). Taken together, these results demonstrate the utility of an in-locus allele switch system for elucidating the in vivo consequences of spontaneous disease-linked mutations.

In humans, *BMPR1A* is known as a causative gene for Juvenile polyposis syndrome (JPS), a condition which predisposes patients to the development of gastrointestinal cancer (*Figure 2A*) (*Howe et al., 2001*; *Sayed et al., 2002*; *Greenman et al., 2007*; *Hardwick et al., 2008*). Although several *BMPR1A* point mutations have been identified in JPS tissue samples, precisely how each lesion influences receptor activity remains unclear. To investigate this, we generated allele switch transgenic strains for five JPS-linked point mutations at sites that are conserved in *Drosophila* Tkv (*Figure 2A*). In order to uniformly induce either hetero- or homozygosity throughout all cells of developing wing discs, larvae of *tkv* allele switch transgenics carrying *hs-flp* were subjected to a prolonged 1 hr heat shock at 72 hr after egg laying (AEL) (*Figure 2B–L*; *Figure 2—figure supplement 1A and B*). We found that these mutations fell into three functional classes: no effect, loss-of-function, and context-dependence. First, although the $BMPR1A^{R443C}$ mutation was detected in JPS patients (*Sayed et al., 2002*; *Greenman et al., 2007*), the corresponding *tkv* mutation produced no obvious phenotypes in either the hetero- or homozygous condition (*Figure 2—figure supplement 1B–H*). In a second allele class, we observed the expected loss-of-function effects. While

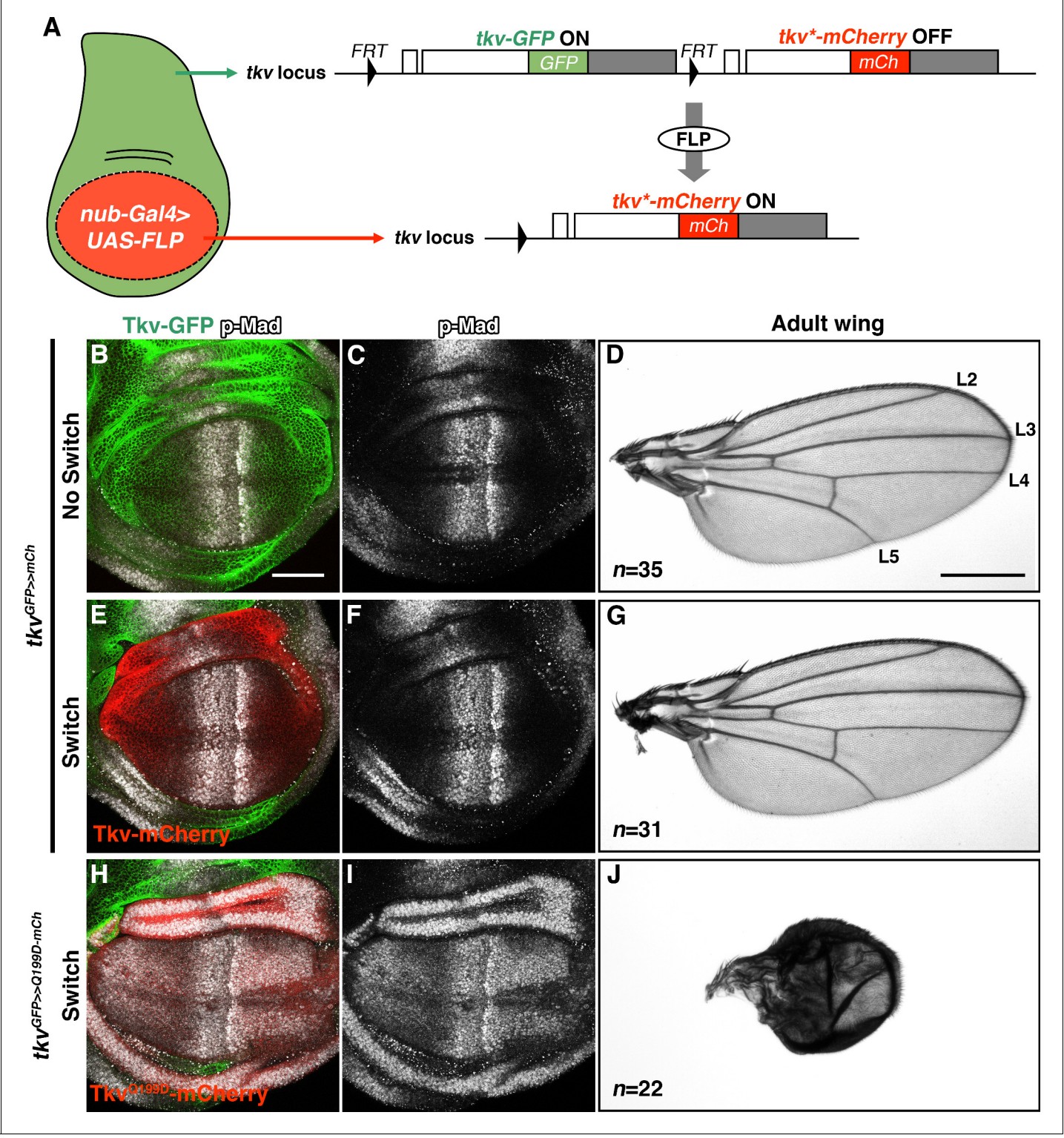

**Figure 1.** Inducible *tkv* allele switch system. (**A**) Schematic illustration of a FLP/*FRT*-mediated *tkv* allele switch. mCherry-tagged Tkv expression was specifically induced in the wing pouch via FLP/*FRT*-mediated recombination that eliminates *tkv-GFP* from the locus. White and grey boxes represent *tkv* coding and 3'UTR sequences, respectively. (**B–G**) *tkv*$^{GFP>>mCherry}$ heterozygous animal before and after allele switch. (**B, C, E, F**) GFP (*green*) and mCherry (*red*) represent endogenous Tkv expression in the developing wing disc. BMP/DPP activity was detected by anti-p-Mad staining (*grey*). (**D, G**) Both Tkv-GFP and Tkv-mCherry expressing animals developed normal adult wings. Longitudinal veins L2-L5 are indicated. (**H–J**) Wing pouch-specific allele switch of the constitutively active form of Tkv, Tkv$^{Q199D}$. Tkv$^{Q199D}$-mCherry (*red*) expression led to ectopic p-Mad activation (*grey*, H, I), resulting in

*Figure 1 continued on next page*

*Figure 1 continued*

wing malformation (J). Scale bars: 50 µm for (B), and 0.5 mm for (D). Anterior is oriented to the left in all wing disc figures and to the top in adult wing images.

DOI: https://doi.org/10.7554/eLife.35258.002

The following figure supplements are available for figure 1:

**Figure supplement 1.** Wing patterning defects caused by *tkv* overexpression.
DOI: https://doi.org/10.7554/eLife.35258.003

**Figure supplement 2.** Spatial regulation of *tkv* allele switch.
DOI: https://doi.org/10.7554/eLife.35258.004

heterozygosity for C40Y, C97R and M442T mutations produced normal wings and similarly mild effects on the p-Mad activity gradient (*Figure 2D–F*; *Figure 2—figure supplement 1A and B*), wing discs homozygous for these mutations displayed distinct phenotypes (*Figure 2I–K*; *Figure 2—figure supplement 1B*). tkv$^{C40Y-mCherry}$ and tkv$^{C97R-mCherry}$ homozygosity resulted in small wing discs with no detectable p-Mad activity, consistent with strong loss-of-function alleles (*Figure 2I and J*; *Figure 2—figure supplement 1B*). In contrast, tkv$^{M442T-mCherry}$ homozygous discs were larger and featured a broadened p-Mad activity gradient (*Figure 2K and M*; *Figure 2—figure supplement 1B*), perhaps due to reduced receptor binding and increased BMP/DPP ligand availability. Supporting this model, occasional residual clones expressing *wild-type* Tkv-GFP exhibited ectopic signaling activity in tkv$^{M442T-mCherry}$ homozygous wing discs (*Figure 2—figure supplement 2*). Similarly, residual *wild-type* clones within tkv$^{C97R-mCherry}$ mutant wing discs showed high levels of BMP activity and frequently produced distinctly rounded outgrowths (*n* = 6/8 discs; *Figure 2N–Q*).

In a third and final allele class, we found that the tkv$^{C90R-mCherry}$ variant possessed distinct signaling capabilities depending on the wing disc location. While tkv$^{C90R-mCherry}$ homozygosity resulted in weaker p-Mad expression in the wing pouch, the same mutation activated BMP signaling in the hinge region in both hetero- and homozygous discs (*Figure 2G,L and M*; *n* = 35 and 32 discs, respectively). Altogether, these results indicate that JPS-associated *tkv* mutations present a variety of wing disc growth phenotypes linked to distinct effects on signaling activity. In addition, we identified an intriguing new allele, tkv$^{C90R}$, which either promotes or represses signaling activity in a position-dependent manner.

Human disease alleles, such as those identified in JPS patients, can be categorized according to either spontaneous or inherited origins. Leveraging the allele switch system, we next sought to model the cellular etiology associated with somatic mutations in JPS-associated alleles (*Figure 3*). Consistent with the results of uniformly-induced tkv$^{C97R}$ heterozygosity during larval development (*Figure 2E*; *Figure 2—figure supplement 1A and B*), germline tkv$^{C97R}$ heterozygotes exhibited little effect on p-Mad expression and developed normal adult wings (*Figure 3A–D and K*). Strikingly, however, when we induced sporadic tkv$^{C97R}$ heterozygous clones at 72 hr AEL, experimental animals showed aberrant p-Mad expression in wing discs and patterning defects in adult wings (*Figure 3E–K*). At the cellular level, clones of tkv$^{C97R}$ heterozygous cells exhibited strongly reduced BMP/DPP signaling activity and increased p-Mad levels in adjacent *wild-type* cells (*Figure 3I and J*). These results demonstrate the unexpectedly detrimental effect of mosaic heterozygosity for a classically recessive allele, an effect we define as *deleterious heteromosaicism*.

To confirm that mosaicism itself was the cause of abnormal wing vein patterning, we more uniformly induced tkv$^{C97R-mCherry}$ heterozygous cells throughout the wing disc by increasing the duration of heat shock from 10 to 60 min. Indeed, longer duration heat shock increased the number of heterozygous cells present in developing discs, and rescued the wing vein phenotypes associated with heterozygous mosaicism (*Figure 3K*). This indicates that deleterious heteromosaic effects derive from the clonal confrontation between heterozygous and *wild-type* cells.

Cell autonomous abrogation of BMP signaling typically leads to clone extrusion and apoptosis (*Adachi-Yamada and O'Connor, 2002*; *Gibson and Perrimon, 2005*; *Shen and Dahmann, 2005*). Consistent with these observations, the majority of homozygous tkv$^{C97R}$ clones were eliminated from the wing epithelia (*Figure 4A*; *Figure 3—figure supplement 1*). However, neither form of cell elimination was observed in wing discs carrying tkv$^{C97R}$ heterozygous clones (*Figure 3F,G,I and J*; *Figure 4B*; *Figure 3—figure supplement 2*). We infer that BMP signal reduction in heterozygous cells was adequate to cause deleterious effects but insufficient to trigger cell removal. Further,

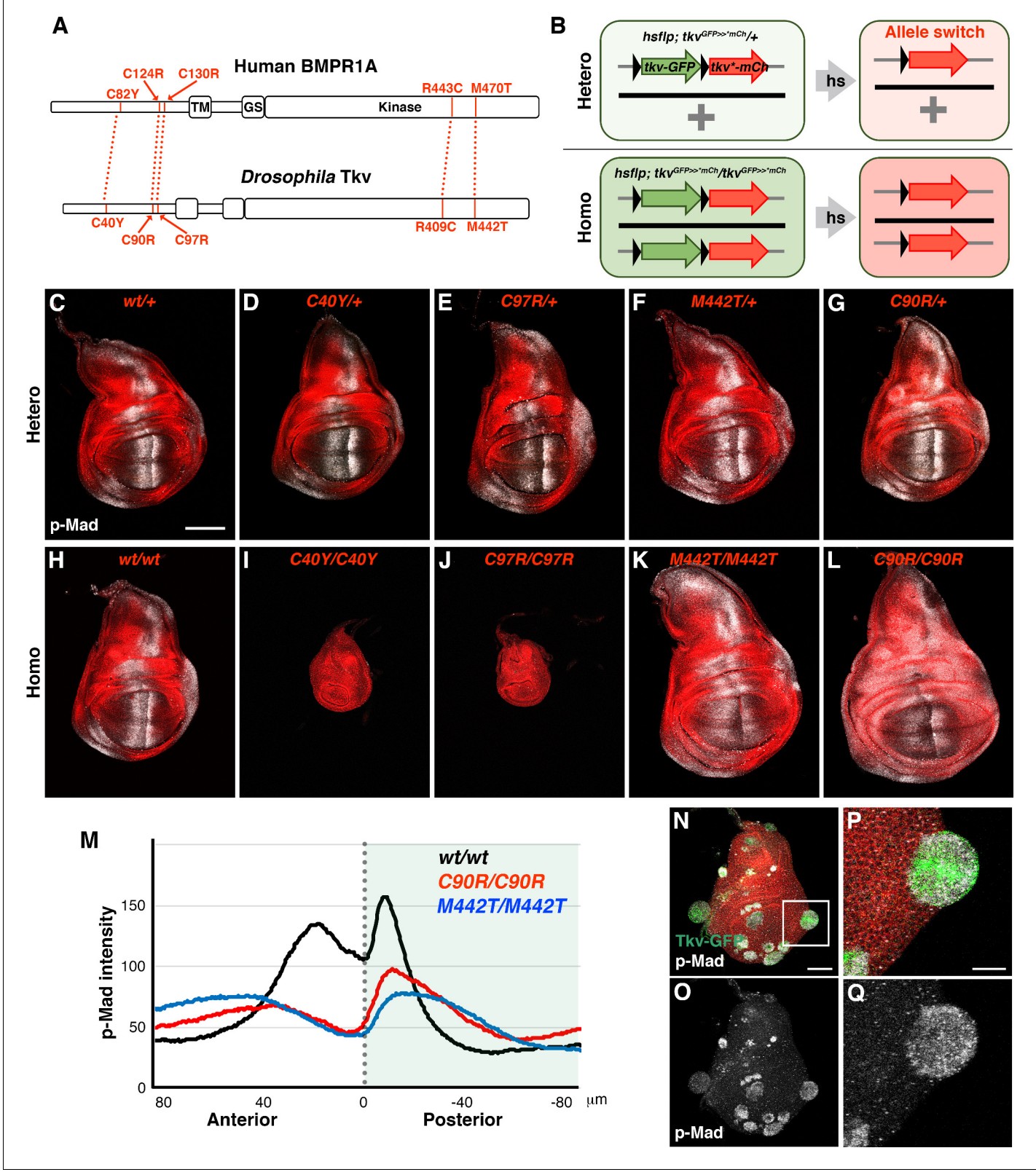

**Figure 2.** Distinct wing disc phenotypes caused by JPS-associated *tkv* mutations. (**A**) Human BMPR1A and *Drosophila* Tkv protein structures. Conserved JPS-associated mutations are indicated. (**B**) Scheme for the induction of *tkv* hetero- and homozygous mutant cells by heat shock. (**C–L**) *wild-type* and mutant forms of Tkv-mCherry (*red*) together with p-Mad (*grey*) expression in either hetero- (**C–G**) or homozygous (**H–L**) wing discs. Scale bars: 100 μm. (**M**) Averaged p-Mad intensity plot profiles for *wild-type tkv^mCherry* (n = 16), *tkv^C90R-mCherry* (n = 15) and *tkv^M442T-mCherry* (n = 16) homozygous wing discs.

*Figure 2 continued on next page*

*Figure 2 continued*

0 indicates the compartment boundary position (posterior is shaded). (N–Q) High levels of BMP activity observed in residual *wild-type* clones within predominantly *tkv*^C97R-mCherry^ homozygous wing discs. Ectopic p-Mad expression (*grey*) is detected in rounded clusters of *wild-type* cells (*green* in N, P). (P and Q) show magnified images of the boxed region in (N). Scale bars: 50 μm for (N), 20 μm for (P). Anterior is oriented to the left side of all images.

DOI: https://doi.org/10.7554/eLife.35258.005

The following source data and figure supplements are available for figure 2:

**Source data 1.** BMP/DPP activation in *wild-type* clones within the *tkv*^C97R^ homozygous background.

DOI: https://doi.org/10.7554/eLife.35258.008

**Figure supplement 1.** Phenotypes of *tkv* mutations.

DOI: https://doi.org/10.7554/eLife.35258.006

**Figure supplement 2.** Strong BMP activation in cells expressing *wild-type* Tkv within the *tkv*^M442T^ background.

DOI: https://doi.org/10.7554/eLife.35258.007

consistent with a general phenomenon not restricted to *tkv*^C97R-mCherry^, heterozygous cell clones for other recessive *tkv* alleles elicited similar phenotypes (*Figure 3—figure supplement 3*). In agreement with previous results (*Figure 2G and L*), we also found that clones heterozygous for *tkv*^C90R-mCherry^ disrupted p-Mad activity in the developing wing blade region but ectopically activated BMP/DPP signaling in the presumptive hinge territory (n = 26/27 discs; *Figure 3—figure supplement 3G–L*). Lastly, we also examined for the effects of deleterious heteromosaicism in other tissues. Although *tkv*^C97R^ heterozygous clones influenced BMP/DPP activity in both eye and haltere discs at the level of Mad phosphorylation, the adult structures developed normally with no obvious defect (*Figure 3—figure supplement 4*). These findings suggest a tissue-specific susceptibility to the same recessive mutations in heterozygous cell clones.

Recent advances in genome-wide association studies report a substantial accumulation of somatic mutations within individuals, resulting in genetic heterogeneity (*Frank, 2014*; *Campbell et al., 2015*; *Fernández et al., 2016*; *Forsberg et al., 2017*; *Machiela and Chanock, 2017*). This mutational load is associated with the initiation and progression of a number of diseases, including cancer and X-linked disorders (*Veltman and Brunner, 2012*; *Deng et al., 2014*; *Fernández et al., 2016*; *Tomasetti et al., 2017*). Here, we have established an inducible allele switch system that allows us to study specific disease-associated point mutations within endogenous loci (*Figures 1* and *2*; *Figure 1—figure supplement 2*; *Figure 2—figure supplements 1* and *2*). For the first time, genome engineering methods make it possible to model the occurrence of spontaneous disease mutations and investigate their biological consequences in living tissue (*Figures 3* and *4*; *Figure 3—figure supplements 2–4*). Using this approach, we have uncovered a unique facet of tissue genetics whereby classically recessive alleles can trigger aberrant development when present in heterozygous cell clones, a phenomenon we call deleterious heteromosaicism (*Figures 3* and *4*; *Figure 3—figure supplements 2* and *3*). Since animals uniformly heterozygous for the same alleles are phenotypically normal, heteromosaic phenotypes likely emerge from local disparities between *wild-type* and heterozygous cells. Cellular heterogeneity associated with homozygous mutant clones is known to perturb tissue integrity by causing cyst formation, cell elimination and apoptosis (*Adachi-Yamada and O'Connor, 2002*; *Gibson and Perrimon, 2005*; *Shen and Dahmann, 2005*; *Hogan et al., 2009*; *Bielmeier et al., 2016*). In contrast, *tkv* deleterious heteromosaic clones showed no evidence of abnormal tissue architecture or cell death (*Figures 3* and *4*; *Figure 3—figure supplements 2* and *3*). Thus, although *tkv* heterozygous cells disrupted normal wing pattern formation, they did not strongly influence growth and appear to escape from homeostatic surveillance mechanisms such as cell competition (*Di Gregorio et al., 2016*).

A prevailing model in cancer genetics, the two-hit hypothesis requires that two independent mutations arise in a single tumor suppressor gene (*Nordling, 1953*; *Knudson, 1971*). Assuming mutagenesis by random DNA replication error, the probability of two independent mutations hitting the same gene within the same cell lineage is relatively low. Here, we report that a single somatic mutation in a putative tumor suppressor causes abnormal development by disrupting the homogeneity of BMPR1A-mediated cellular communication at the tissue level (*Figures 3* and *4*; *Figure 3—figure supplements 2* and *3*). Given that a number of disease-linked mutations are known in major signaling pathway components, we speculate that deleterious heteromosaicism is not limited to the BMP signaling pathway. Further, we propose that investigation into the unique facets of tissue

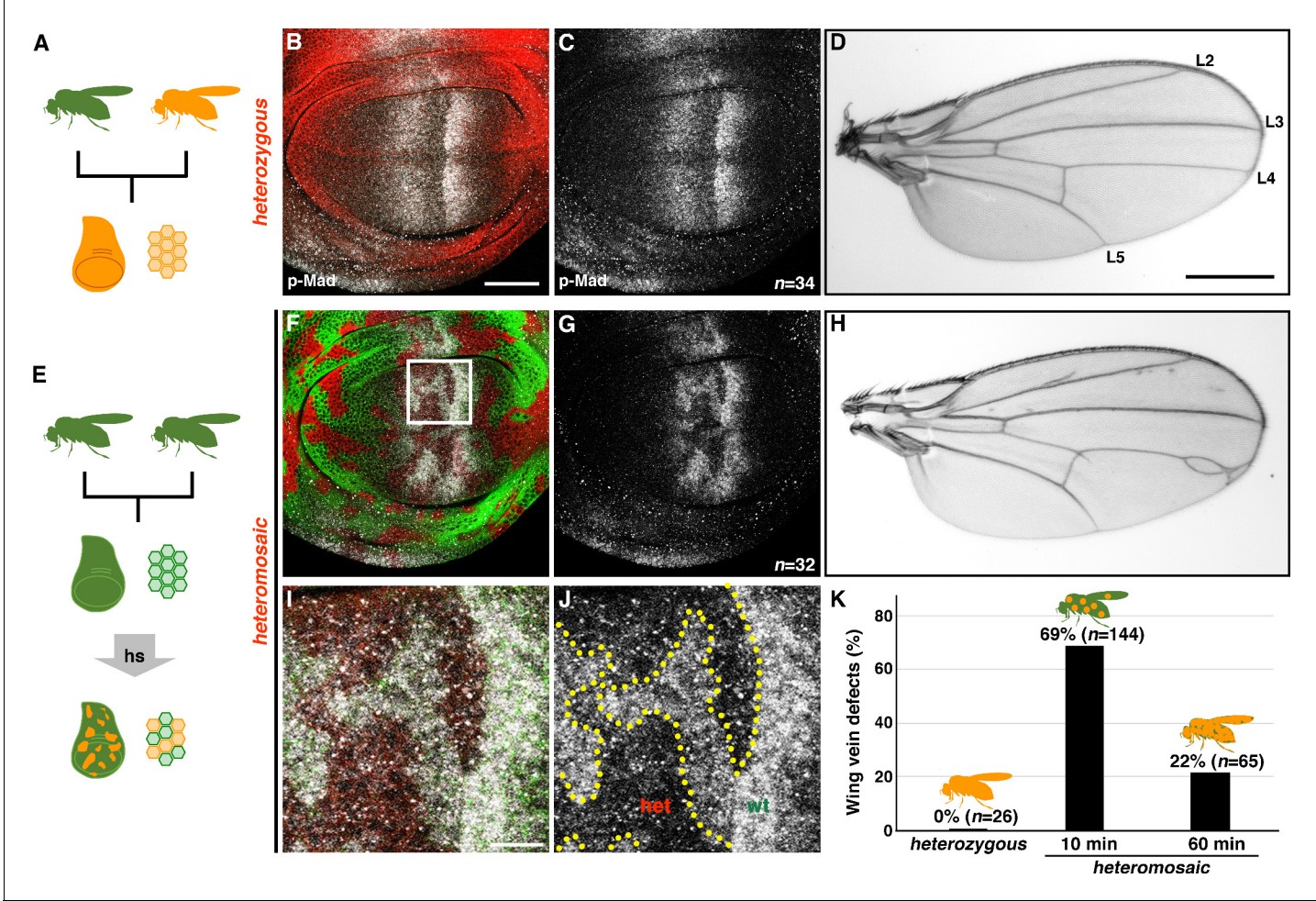

**Figure 3.** *tkv* heterozygous mosaicism disrupts wing pattern formation. (**A–D**) Germline inheritance of the recessive *tkv*^C97R-mCherry^ mutation (**A**). Tkv^C97R^-mCherry (*red*) and p-Mad (*grey*) expression in a heterozygous wing disc (**B, C**). *tkv*^C97R^ heterozygotes developed normal wings (**D**). Positions of longitudinal vein L2-L5 are shown. (**E–J**) Generating somatic clones of *tkv*^C97R^ heterozygous cells by heat shock (**E**). *wild-type* and *tkv*^C97^ heterozygous cells are shown by *green* and *red*, respectively (**F, G, I, J**). *tkv*^C97R-mCherry^ heteromosaicism disrupted the p-Mad activity gradient (*grey*) and caused wing vein patterning defects (**H**). (**I, J**) Magnified images of the boxed area in (**F**). A similar effect on p-Mad activity was observed in male wing discs, although adult males exhibited milder wing phenotypes. Scale bars: 50 μm for (**B**), 0.5 mm for (**D**), and 10 μm for (**I**). Anterior is oriented to the left side of wing disc images and to the top side of adult wing pictures. (**K**) Quantification of adult wing phenotypes in *tkv*^C97R-mCherry^ heterozygous and heteromosaic animals. Longer heat shock generated more uniformly heterozygous cell populations and reduced mosaicism, rescuing wing vein phenotypes.

DOI: https://doi.org/10.7554/eLife.35258.009

The following figure supplements are available for figure 3:

**Figure supplement 1.** Basal extrusion of *tkv*^C97R^ homozygous mutant clones from the wing epithelia.
DOI: https://doi.org/10.7554/eLife.35258.010

**Figure supplement 2.** *tkv* heteromosaicism does not induce cell death.
DOI: https://doi.org/10.7554/eLife.35258.011

**Figure supplement 3.** Additional JPS-associated *tkv* mutations exhibit deleterious heteromosaicism.
DOI: https://doi.org/10.7554/eLife.35258.012

**Figure supplement 4.** Effects of *tkv*^C97R^ heterozygous mutant clones in the developing eye and haltere discs.
DOI: https://doi.org/10.7554/eLife.35258.013

genetics has the potential to shed new light on the mechanisms of how disease-linked recessive mutations impact human health.

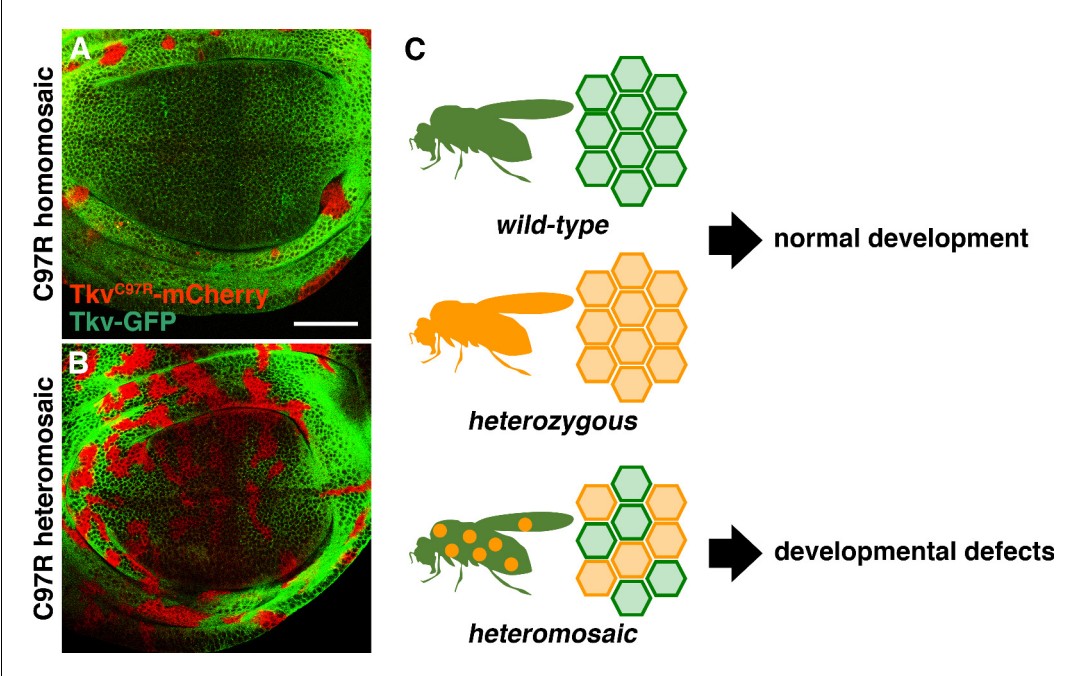

**Figure 4.** Deleterious heteromosaicism. (**A, B**) *tkv*[C97R-mCherry] homozygous clones were eliminated from wing disc epithelia (**A**), while heterozygous cells were retained (**B**). Scale bars: 50 µm. Anterior is to the left. (**C**) While *wild-type* animals and heterozygous carriers both develop normally, animals carrying heterozygous somatic clones exhibit developmental defects.

DOI: https://doi.org/10.7554/eLife.35258.014

## Materials and methods

### Generation of *tkv* allele switch transgenic lines

Two *tkv* sgRNA DNA constructs were generated using the primers listed in *Supplementary file 1*. *tkv* sgRNA DNA construct 1 (primers 1 and 2), and *tkv* sgRNA DNA construct 2 (primers 3 and 4), were annealed and cloned into the BbsI site of *pBFv-U6.2* (*Kondo and Ueda, 2013*). A ubi-mCherry selection marker was generated as previously described (*Akiyama and Gibson, 2015a*). To obtain a donor DNA construct for generating a *tkv* allele switch founder transgenic line, six PCR fragments were prepared using primers 5 to 16 in *Supplementary file 1*. The PCR products were combined via Gibson assembly (NEB).

A DNA mixture containing two *tkv* sgRNA DNAs and the donor plasmid (250 ng/µl for each) was injected into the posterior side of embryos expressing Cas9 controlled by the *nos* promoter (*Kondo and Ueda, 2013*). Transgenic flies were selected by mCherry expression and further confirmed by DNA sequencing. Finally, the selection cassette was removed via Cre/*loxP*-mediated recombination, resulting in the *tkv allele switch founder* (see *Figure 1—figure supplement 2A*).

To generate *tkv* allele cassettes, a *tkv-mCherry* DNA fragment with 5' EcoRI and 3' AscI sites was constructed by Gibson assembly using primers 17 to 22 in *Supplementary file 1*. After Gibson assembly, a second round of PCR was conducted using primers 17 and 22 (*Supplementary file 1*). The resulting PCR product was cloned into *pCRII blunt TOPO* (Thermo Fisher Scientific).

Each *tkv* point mutation was induced with the Q5 Site-Directed Mutagenesis Kit (NEB), using primers 23 to 34 described in *Supplementary file 1*. After DNA sequencing, *pCRII blunt TOPO tkv\*-mCherry* DNAs were digested by EcoRI and AscI, and cloned into *w + attB* (addgene #30326), resulting in *w + attB tkv\*-mCherry*.

To establish *tkv\** allele switch transgenic lines, *w + attB tkv\*-mCherry* DNA constructs were injected into the posterior regions of embryos obtained from a cross of *nos-phiC31 int. NLS* (on X, Bloomington #34770) and the *w; tkv allele switch founder*. Transformants were screened by the presence of red eye color (see *Figure 1—figure supplement 2A*).

## Heat shock

To mimic induction of somatic mutation in vivo, FLP expression was induced at 37°C for 10 (*Figure 3F–J*; *Figure 3—figure supplements 1–4*), 15 (*Figure 2N–Q*), 30 (*Figure 2—figure supplement 2*), or 60 (*Figure 2C–L*; *Figure 2—figure supplement 1A and B*) minutes at 72 hr AEL.

*tkv\** allele switch heterozygous cross (*Figure 2C–G*: *Figure 3F–K*; *Figure 2—figure supplement 1A and B*; *Figure 3—figure supplements 2–4*): w, hs-flp x w; tkv\* allele switch/CyO tkv allele switch homozygous cross (*Figure 2H–Q*; *Figure 2—figure supplement 1B*; *Figure 2—figure supplement 2*): w, hs-flp; tkv\* allele switch/CyO,sChFP x w, hs-flp/Y; tkv\* allele switch/CyO,sChFP

*tkv* allele switch in heterozygous backgrounds (*Figure 4A and B*; *Figure 3—figure supplement 1*): w; tkv\*-mCherry/CyO,twi-Gal4, UAS-GFP x w, hs-flp/Y; tkv\* allele switch/CyO,sChFP

## Gal4/UAS

For region-specific *tkv* allele switching, FLP expression was induced using *nub-GAL4* (*Calleja et al., 1996*), *hh-Gal4* (*Mullor and Guerrero, 2000*), and *ap-GAL4* (*Calleja et al., 1996*) for FLP/*FRT*-mediated recombination.

## nub-GAL4

*tkv* allele switch cross: *w; nub-Gal4/CyO; UAS-FLP/TM6C x w; tkv*$^{GFP>>mCherry}$*/CyO* (*Figure 1E–G*; *Figure 1—figure supplement 2E*) *tkv*$^{Q199D}$ allele switch cross: *w; nub-Gal4/CyO; UAS-FLP/TM6C x w; tkv*$^{GFP>>Q199D-mCherry}$*/CyO* (*Figure 1H–J*)

## hh-GAL4

*w; hh-Gal4,UAS-FLP/TM6B x w; tkv*$^{GFP>>mCherry}$*/CyO* (*Figure 1—figure supplement 2C*)

## ap-GAL4

*w; ap-Gal4/CyO; UAS-FLP/TM6C x w; tkv*$^{GFP>>mCherry}$*/CyO* (*Figure 1—figure supplement 2D*)

## Immunohistochemistry and imaging

Third-instar wing discs were dissected from female larvae and stained with rabbit anti-pSmad3 (EP823Y, 1:1,000, abcam) (*Akiyama et al., 2012*; *Akiyama and Gibson, 2015a*) and rabbit anti-cleaved *Drosophila* Dcp-1 (Asp216, 1:500, Cell Signaling Technology) (*Harris et al., 2016*; *Verghese and Su, 2016*). Alexa conjugated secondary antibodies (1:500, Invitrogen) were used to detect primary antibodies. Can Get Signal Immunostain Solution B (TOYOBO, New York, NY) was used to dilute primary antibodies. Images of wing discs and adult female wings were obtained using a Leica TCS SP5 confocal microscope and a Leica CTR 5000, respectively.

## Image analysis

To generate p-Mad intensity plot profiles, all images were collected at the same confocal setting and analyzed using the RGB profiler of FIJI. The 'Measure' function of FIJI was used to analyze sizes of wing discs.

## Acknowledgements

We thank A Ikmi for extensive discussion and suggestions, and H Nakato and the Bloomington Stock Center for fly stocks. We thank W Fropf, L Ellington and E Bauerly for a critical reading of the manuscript, H Yin for administrative support, and members of the Gibson lab for discussions and advice. This work was supported by funding from the Stowers Institute for Medical Research and NIH R01 GM111733 to MCG.

## Additional information

### Funding

| Funder | Grant reference number | Author |
|---|---|---|
| Stowers Institute for Medical Research | | Matthew C Gibson |
| National Institutes of Health | GM111733 | Matthew C Gibson |

The funders had no role in study design, data collection and interpretation, or the decision to submit the work for publication.

### Author contributions

Takuya Akiyama, Conceptualization, Investigation, Writing—original draft, Writing—review and editing; Sırma D User, Investigation; Matthew C Gibson, Conceptualization, Funding acquisition, Investigation, Writing—original draft, Writing—review and editing

### Author ORCIDs

Takuya Akiyama (iD) http://orcid.org/0000-0002-9291-0620
Matthew C Gibson (iD) http://orcid.org/0000-0001-5588-8842

### Decision letter and Author response

Decision letter https://doi.org/10.7554/eLife.35258.020
Author response https://doi.org/10.7554/eLife.35258.021

## Additional files

### Supplementary files

• Supplementary file 1. Primers used in this study. EcoRI and AscI sites are underlined in 17 and 22, respectively. Lowercase characters indicate point mutations.
DOI: https://doi.org/10.7554/eLife.35258.015

• Transparent reporting form
DOI: https://doi.org/10.7554/eLife.35258.016

### Data availability

All primary data is available at the Stowers Institute Original Data Repository (https://www.stowers.org/research/publications/libpb-1261)

The following dataset was generated:

| Author(s) | Year | Dataset title | Dataset URL | Database, license, and accessibility information |
|---|---|---|---|---|
| Akiyama T, User S, Gibson M | 2018 | LIBPB-1261 | https://www.stowers.org/research/publications/libpb-1261 | Available at the Stowers Institute Original Data Repository |

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
