## [Decision Letter]

Thank you for submitting your article "Deleterious heteromosaic phenotypes in somatic clones heterozygous for recessive *BMPR1A* disease alleles" for consideration by *eLife*. Your article has been reviewed by three peer reviewers, one of whom, Jody Rosenblatt, is a member of our Board of Reviewing Editors and the evaluation has been overseen by Didier Stainier as the Senior Editor.

The reviewers have discussed the reviews with one another and the Reviewing Editor has drafted this decision to help you prepare a revised submission.

Summary:

As you will see all three reviewers found your manuscript conceptually very novel and interesting and are in favor of its publication. This manuscript identifies a new genetic phenomenon occurring in clones of heterozygous mutant cells when surrounded by wild-type cells, suggesting a new mechanism for maintaining altered, potentially diseased tissue, rather than previous models implicating loss of heterozygosity. While there was some discussion that it would be interesting to reveal the mechanism behind heterozygous tissue giving pattern defects, there was an overall impression that discovery of the phenomenon was noteworthy. Additionally, as the authors have generated very useful new fly lines in order to perform these elegant experiments, the lines will be helpful for finding other cases for this phenomenon. As a novel concept, this is certainly worth publishing. There were only a couple of points that we felt should be addressed before publication.

Essential revisions:

1) The use of human disease-causing mutations makes an important link with Juvenile Polyposis, helping to underscore the functional importance of a subset of these mutations. For this reason, it would be interesting to try the tkv switch experiment in another tissue, for example in the fly intestine, where tkv RNAi or mutant clones seem to have strong phenotypes. (https://elifesciences.org/articles/01857/figures).

2) As the main proposal for this mechanism is that small patches of homozygous mutants would be competed out, whereas heterozygous cells would not be, a figure comparing these results with the same allele would be clarifying, especially when used as a figure to demonstrate the concept.

---

## [Author Response]

Essential revisions:1) The use of human disease-causing mutations makes an important link with Juvenile Polyposis, helping to underscore the functional importance of a subset of these mutations. For this reason, it would be interesting to try the tkv switch experiment in another tissue, for example in the fly intestine, where tkv RNAi or mutant clones seem to have strong phenotypes. (https://elifesciences.org/articles/01857/figures).

Although we tried adult midgut experiments, we could not clearly visualize Tkv due to its low level of endogenous expression in the gut. Therefore, we focused on the effect of *tkv* allele switching in other tissues. We found that while *tkv^C97R^* heterozygous clones clearly impact BMP/DPP activity in both eye and haltere discs, these structures nonetheless developed normally (Figure 3—figure supplement 4). These findings indicate a tissue-specific susceptibility to heterozygous cell clones, which we now discuss in the revised manuscript.

2) As the main proposal for this mechanism is that small patches of homozygous mutants would be competed out, whereas heterozygous cells would not be, a figure comparing these results with the same allele would be clarifying, especially when used as a figure to demonstrate the concept.

As suggested by the reviewer,we now show that *tkv^C97R^* homozygous cells are eliminated from wing epithelia (Figure 4; Figure 3—figure supplement 1).